# Histone Acetyltransferase GCN5 Affects Auxin Transport during Root Growth by Modulating Histone Acetylation and Gene Expression of PINs

**DOI:** 10.3390/plants11243572

**Published:** 2022-12-17

**Authors:** Stylianos Poulios, Foteini Tsilimigka, Areti Mallioura, Dimitris Pappas, Eleftheria Seira, Konstantinos Vlachonasios

**Affiliations:** 1Department of Botany, School of Biology, Faculty of Science, Aristotle University of Thessaloniki, 54124 Thessaloniki, Greece; 2Postgraduate Program Studies “Applications of Biology—Biotechnology, Molecular and Microbial Analysis of Food and Products”, School of Biology, Faculty of Science, Aristotle University of Thessaloniki, 54124 Thessaloniki, Greece; 3Natural Products Research Centre of Excellence (NatPro-AUTh), Center of Interdisciplinary Research and Innovation of Aristotle University of Thessaloniki (CIRI-AUTh), 57001 Thessaloniki, Greece

**Keywords:** ADA2b, root meristem, histone acetylation, GCN5, PIN1, PIN3, PIN4, root growth, auxin transport, gene expression

## Abstract

General Control Non-Derepressible 5 (GCN5) is a histone acetyltransferase that targets multiple genes and is essential for the acetylation of Lysine residues in the N-terminal tail of histone H3 in Arabidopsis. GCN5 interacts with the transcriptional coactivator Alteration/Deficiency in Activation 2b (ADA2b), which enhances its activity functioning in multiprotein complexes, such as the Spt-Ada-Gcn5-Acetyltransferase complex (SAGA). Mutations in *GCN5* and *ADA2b* result in pleiotropic phenotypes, including alterations in the growth of roots. Auxin is known to regulate root development by modulating gene expression patterns. Auxin moves polarly during plant growth via the Pin-formed (PIN) auxin efflux transport proteins. The effect of GCN5 and ADA2b on auxin distribution at different stages of early root growth (4 to 7 days post-germination) was studied using the reporter lines *DR5rev::GFP* and *PIN1::PIN1-GFP*. In wild-type plants, auxin efflux transporter PIN1 expression increases from the fourth to the seventh day of root growth. The PIN1 expression was reduced in the roots of *gcn5-1* and *ada2b-1* compared to the wild type. The expression of PIN1 in *ada2b-1* mutants is confined only to the meristematic zone, specifically in the stele cells, whereas it is almost abolished in the elongation zone. Gene expression analysis showed that genes associated with auxin transport, *PIN1*, *PIN3* and *PIN4*, are downregulated in *gcn5-1* and *ada2b-1* mutants relative to the wild type. As a result, auxin accumulation was also reduced in *gcn5-1* and *ada2b-1* compared to wild-type roots. Furthermore, acetylation of Lysine 14 of histone H3 (H3K14) was also affected in the promoter and coding region of *PIN1*, *PIN3* and *PIN4* genes during root growth of Arabidopsis in *gcn5* mutants. In conclusion, GCN5 acts as a positive regulator of auxin distribution in early root growth by modulating histone H3 acetylation and the expression of auxin efflux transport genes.

## 1. Introduction

The plant hormone auxin affects multiple developmental processes, such as embryogenesis, organogenesis, cell elongation, division, and differentiation and responses to environmental stimuli [1,2]. Auxin is unique among plant hormones because it has a directional, cell-to-cell transport implemented by membrane-bound protein efflux transporters [2,3]. This directional auxin transport is regulated by the asymmetric localization of the auxin efflux transporter of the PIN family [4,5,6]. In *Arabidopsis thaliana*, the *PIN* gene family has eight members [7], divided into two subfamilies. The first includes PIN1-4, 6, and PIN7 are the long or canonical plasma membrane-localized efflux carriers, whereas the second has PIN5 and PIN8 are the short or non-canonical endoplasmic reticulum-localized auxin carriers [7,8]. The activity and localization of PINs are regulated and fine-tuned by internal and external signals at many different levels, including transcriptional and posttranscriptional regulation, sub-cellular trafficking, and protein turnover [2,6].

Auxin has a significant role in root development. Early experiments have shown that auxin can both stimulate root initiation and inhibit root growth, depending on the concentration [9]. Mutations in the auxin transporters of the PIN family result in plants with aberrant root development, suggesting the important role of auxin flow for normal root growth [10,11]. Auxin is transported from the shoot, towards the tip of the root (rootward) with the help of PIN transporters. There, it is laterally redistributed in the root cap to the outer root cell types, then, auxin is transported back to shoot again via PIN proteins [9]. This process creates an auxin gradient with a maxima at the QC [12]. This auxin gradient is thought to be interpreted by the Plethora (PLT) transcription factors [13] that themselves regulate auxin distribution by suppressing the expression of a subset of PINs [14]. Auxin also interacts with other phytohormones, especially cytokinin with which is thought to have an antagonistic relationship (reviewed in [9,15]).

GCN5 (also known as HAG1) is a histone acetyltransferase [16,17] involved in many developmental processes and responses to environmental stimuli [18,19]. The *gcn5* mutants exhibit pleiotropic phenotypes, including dwarfism, loss of apical dominance, upward curled and serrated leaves, abnormal inflorescence meristem, abnormal flower development, and shorter roots [18,20,21,22]. In the root precisely, GCN5 has been shown to promote the expression of the PLT transcription factors that regulate root stem cell niche specification [10,13,22]. In addition, GCN5 promotes cytokinin signaling, especially in the columella and lateral root cap cells [23]. GCN5 is the catalytic subunit of the Arabidopsis SAGA (Spt-Ada-Gcn5-Acetyltransferase) complex [18,24] and has been shown to interact with the transcriptional adaptors ADA2a and ADA2b [16,25,26]. The *ada2a* mutants have no visible phenotype [27]. On the contrary, *ada2b* mutants present a phenotype highly similar but not identical to *gcn5* mutants [20,27,28]. The *ada2b* phenotype includes dwarfism, small dark green curled leaves, infertility, and a shorter root compared to both the wild type and *gcn5* seedlings, a result of both smaller cells in the elongation zone and a smaller meristem zone [22]. As was the case for GCN5, ADA2b also promotes cytokinin signaling in the root cap of Arabidopsis [23]. 

Auxin plays a pivotal role during adventitious root formation [29]. In tree species, often an exogenous application of auxin is required for rooting. Adventitious root formation is a key step in vegetative propagation of elite germplasm, since the recalcitrance of many trees becomes a limiting factor in horticulture and forestry [30]. Therefore, it is crucial to better our understanding of the hormonal and molecular mechanisms of auxin function. In this study, we examine the roles of GCN5 and ADA2b on the polarized transport of auxin during early root growth of Arabidopsis seedlings. We utilized genetic and molecular approaches to show that GCN5 and ADA2b affect the expression of several PIN genes, possibly by modulating histone acetylation in their genomic loci in this developmental stage.

## 2. Results

### 2.1. The Histone Acetyltransferase GCN5 Affects Auxin Transport during Early Root Growth

In previous work we have shown that histone acetyltransferase GCN5 and the associated transcriptional adaptor ADA2b, are necessary for auxin-induced cytokinin signaling during early root growth [23]. Since auxin is directionally transported to the sites needed by the action of the PIN family of auxin efflux carriers [1,4] and GCN5 has been shown to promote the expression of PIN1 auxin transporter during gynoecium development [31], we hypothesized that GCN5 could also affect PIN1 expression during early root growth. To test this hypothesis, we crossed the *gcn5-1* mutant with the translational fusion reporter line *PIN1:PIN1-GFP* [31,32] and monitored PIN1-GFP expression from the fourth to the seventh day after germination. The PIN1 pattern of expression has been previously described [33,34,35]. PIN1 is basally localized in the stele cells in the root meristematic zone (Figure 1A,D). PIN1 is detected at the pericycle and the endodermis, as well as in the quiescent center (QC) and the four cortex and epidermis initial cells (Figure 1A, D). In wild-type plants, the GFP signal, indicating the amount of PIN1 protein, increases from the fourth to the seventh day (Figure 1A). On the seventh day, the signal was detected from the QC towards the central cylinder of the root (Figure 1A). The signal had a stable intensity along the meristematic zone up to the elongation zone (Figure 1A). In the *gcn5-1* mutants, the GFP signal is significantly reduced (Figure 1B,D). Interestingly, the GFP signal was not increased in the *gcn5-1* mutant from the fourth day to the seventh day of root growth (Figure 1B) but in fact it was dwindling with time (Figure 1C). A quantification analysis of the fluorescence signal confirms that the GFP was reduced significantly in *gcn5-1* roots (Figure 1C,D). These results suggest that GCN5 is a positive regulator of auxin transport during root growth, having a synergistic positive effect on the expression of PIN1 transporter during root growth.

### 2.2. ADA2b Affects Auxin Transport during Early Root Growth

To test if the GCN5-associated protein ADA2b also affects PIN1 expression, we crossed *ada2b-1* mutants with the *PIN1:PIN1-GFP* line. Similarly, to GCN5, ADA2b also promotes PIN1 expression during root growth. The GFP signal is significantly reduced in *ada2b-1* mutants compared to the wild type on all days examined (Figure 2A,B,D). In *ada2b-1* mutants, not only the levels of expression of PIN1 are reduced but also the expression pattern is limited (Figure 2D). The GFP signal is detected only in the central stele cells and is absent from the pericycle and endodermis cells (Figure 2D). The expression of PIN1 in *ada2b-1* mutants is also confined only to the meristematic zone and is almost abolished in the elongation zone (Figure 2A,B,D). Quantifying the signal showed that it was significantly reduced in the *ada2b-1* compared to the wild type in all four days. Contrary to *gcn5-1* mutants, PIN1 expression in *ada2b-1* mutants was increased from the fourth to seventh days of root growth (Figure 2C). These results show that the transcriptional adaptor ADA2b also functions as a positive regulator of auxin flow by enhancing the expression of PIN1 during root growth.

### 2.3. The Histone Acetyltransferase GCN5 and the Transcriptional Adaptor ADA2b Promote the Expression of PINs in the Root of Arabidopsis

The role of GCN5 and ADA2b on *PIN* family gene expression was monitored in the roots of seven-day-old plants. *PIN1*, *PIN3* and *PIN4* gene expression are downregulated by almost 50% in the roots of both *gcn5-1* and *ada2b-1* mutants compared to wild-type plants (Figure 3A,C,D). In contrast, *PIN2* and *PIN7* expression showed no significant change in their levels (Figure 3B,E), suggesting that GCN5/ADA2b’s effect on the expression of PINs is gene-specific.

### 2.4. GCN5 and ADA2b Affect Auxin Distribution at the Root Tip during Early Root Growth

To test if the changes in the expression of PINs observed in *gcn5-1* and *ada2b-1* mutants result in altered auxin distribution in the root tip, we monitored the expression of the auxin-responsive *DR5rev::GFP* reporter line in the gcn5 and *ada2b* background from the fourth to the seventh day after germination. On the fourth and fifth days after germination, the *DRrev::GFP* signal in *gcn5-1* and *ada2b-1* was mostly similar to the wild-type plants (Figure 4A–C). On the sixth day after germination, *gcn5* and *ada2b* display altered distribution of the GFP signal; the signal becomes more intense in the QC and less intense in the columella cells (Figure 4A,B). On the seventh day, the changes in the DR5 signal become even more apparent, with a significant enhancement of the signal in the QC and a great reduction in the columella cells (Figure 4A–C). The differences in *DR5* expression are more pronounced in the *ada2b-1* mutant than in the *gcn5-1* (Figure 4A–C). In conclusion, these results suggest that changes in the expression of auxin transport genes observed in *gcn5* and *ada2b* mutants lead to changes in the distribution of auxin in the root tip. 

### 2.5. GCN5 and ADA2b Affect the Expression of Genes Involved in Auxin Biosynthesis and Signaling

The reduced levels of auxin distribution in the root of *gcn5-1* and *ada2b-1* mutants suggest that the transcriptional adaptors GCN5 and ADA2b may also affect, besides auxin transport, auxin biosynthesis and signaling. To further characterize the molecular role of GCN5 and ADA2b in the auxin pathway, the expression levels of auxin-related genes in roots of seven-day-old wild-type, *gcn5-1* and *ada2b-1* plants were analyzed.

Specifically, *YUCCA 5*, *8* and *9* were studied since their expression was reported in the root cap cells [35]. The expression of *YUCCA 5* was only increased in *ada2b-1*, in comparison to *gcn5-1* and wild-type plants. *YUCCA 8* was increased, while *YUCCA 9* was decreased in both *gcn5-1* and *ada2b-1* relative to the wild type. (Figure 5A–C). The expression of *TAA1*, the first enzyme of the auxin biosynthesis pathway [36], was reduced in *ada2b-1* but not in *gcn5-1* mutants, as the reduction was statistically significant only for *ada2b-1* (Figure 5D). Therefore, we conclude that GCN5 and ADA2b regulate the auxin biosynthesis pathway in the root. The *GH3.3* gene is critical for auxin homeostasis [37]; in both *gcn5-1* and *ada2b-1 GH3.3* was overexpressed relative to the wild-type (Figure 5F), indicating root auxin availability in *gcn5-1* and *ada2b-1* may be reduced. In addition, the *IAA3* gene, which represses auxin signaling [38], was downregulated in the *gcn5-1* and *ada2b-1* mutants (Figure 5E), indicating that auxin responses were also reduced in both *gcn5-1* and *ada2b-1* mutants.

### 2.6. GCN5 and ADA2b Affect Histone H3K14 Acetylation on PIN Genomic Loci

To examine whether the observed altered expression of *PIN* genes in the *gcn5-1* and *ada2b-1* single mutants could result from altered histone acetylation of their loci, we performed ChIP analysis in the roots of seven-day-old seedlings using antibodies against histone H3 acetylated lysine 14 (H3K14Ac) and non-acetylated histone H3. H3K14 is the GCN5 target for acetylation [39,40]. For each genomic locus, two regions were examined, the promoter region, spanning from the transcription start site (+1) roughly up to about 200 bases upstream (−200), and an additional region inside the second exon. At the promoter region of the PIN1 locus, H3K14 acetylation does not change in the *gcn5-1* roots, whereas it is increased in *ada2b-1* roots. In the coding region of *PIN1*, the *gcn5-1* mutants display significantly reduced H3K14 acetylation compared to the wild type and *ada2b* mutants (Figure 6A). A similar pattern is exhibited in the *PIN3* locus; *gcn5-1* mutants have reduced H3K14 acetylation in the coding region but not in the promoter. *Ada2b-1* shows increased acetylation in both regions compared to wild-type roots (Figure 6B). In *the PIN4* locus, *gcn5-1* has reduced H3K14 acetylation in both the promoter and coding region. In contrast, *ada2b-1* displays increased H3K14 acetylation in both regions (Figure 6C). These results suggest that GCN5 is a positive modulator of histone H3K14 acetylation on the genomic loci of *PIN* genes. Furthermore, ADA2b is a positive regulator of *PIN* gene expression by a different mechanism than GCN5. 

## 3. Discussion

Herein, we explored the role of histone acetyltransferase GCN5 and the transcriptional adaptor ADA2b in auxin flow during early root growth. We found that GCN5 and ADA2b regulate early root growth in Arabidopsis by modulating the expression and histone acetylation of members of the PIN family of auxin efflux transporters.

Auxins are involved in various developmental responses in plants, including root apical meristem formation and maintenance [41]. The auxin gradient is established and maintained by auxin transport proteins that are essential for root organogenesis [32,33]. Auxin concentration along the roots arises from collective activities and topology of the PIN proteins, the AUX1/LAX family proteins [10,42,43], and the multidrug-resistant/P-glycoprotein family proteins [44].

The roots of *gcn5-1* and *ada2b-1* mutants at the beginning of growth show the same levels of auxin signaling as the wild-type plants, implying that auxin is present during the embryogenesis of these plants. However, during root growth, auxin levels, as monitored by the DR5 reporter, decrease and accumulate in the QC [45]. The auxin accumulation in the QC observed in *gcn5,* and *ada2b* mutants could arise for various reasons.

First, fewer auxin molecules could be biosynthesized in the cells produced during root growth. This scenario is supported by the reduced expression of *TAA1* and *YUCCA9* in the *gcn5-1* and *ada2b-1* mutants. Therefore, GCN5 and ADA2b are important for proper auxin biosynthesis during root growth.

Second, as root growth proceeds, the initial amount of auxin produced during embryogenesis in *gcn5-1* and *ada2b-1* is reduced relative to the wild-type. This could be an effect of increased auxin degradation/inactivation in the mutants. Indeed, the *GH3.3* auxin conjugation gene is overexpressed in *gcn5-1* and *ada2b-1* [this study, 22], indicating increased auxin inactivation. Thus, in the absence of both GCN5 and ADA2b, auxin degradation is promoted. However, beyond GCN5 and ADA2b there are other component(s) implicated in auxin biosynthesis and regulation that await to be unveiled.

Third, our data indicate that GCN5 and ADA2b affect auxin transport. The root cap cells affect the availability and distribution of auxin in the root meristem by affecting auxin transport through PIN proteins and, ultimately, the location of the auxin minimum and transition zone [46]. Indeed, the auxin efflux transporter PIN1 is expressed in the meristematic zone at a lower intensity in *gcn5-1* and *ada2b-1* mutants than in the wild type. In *ada2b-1,* PIN1 expression disappears from the pericycle and endodermis cell layers. Therefore, our data indicate that GCN5 and ADA2b are positive regulators of PIN1 expression in the early days of root growth. Furthermore, assuming that the initial available amount of auxin is catabolized in the cap cells, then because of reduced auxin transport through PIN1, it is impossible to re-infuse the cells with auxin, resulting in reduced auxin signaling. Gene expression experiments also support the latter. The expression of *PIN1*, *PIN3* and *PIN4* is reduced in *gcn5-1* and *ada2b-1* compared to the wild type.

In contrast, *PIN2* expression is not affected; thus, auxin is typically transported from the root tip to the elongation zone in *gcn5-1* and *ada2b-1* mutants. These results suggest that GCN5 and ADA2b contribute to the proper transport of auxin through the central cylinder to the root tip and the QC, resulting in a sufficient amount of auxin being channeled for root growth. Therefore, the absence of GCN5 and ADA2b leads to disruption of auxin transport, which ultimately accounts for the reduced root size observed in the *gcn5-1* and *ada2b-1* mutants. A similar mechanism has been found to occur in Arabidopsis gynoecium. Specifically, GCN5 and ADA2b contribute to proper female development by positively regulating auxin transport through PIN1 [31]. However, locally, auxin biosynthesis in the QC is necessary and sufficient for the maintenance of the root meristem when the auxin polar transport system is affected [47].

Auxin regulates QC’s maintenance and the root meristem’s activity through PLETHORA transcription factors (PLTs) [14]. GCN5 and ADA2b regulate stem cell maintenance by regulating *PLT* expression, while PLT2 overexpression reverts the defective stem cell phenotype in *gcn5-1* and *ada2b-1* [22]. Expression of PLTs is induced by PIN-driven auxin gradient. Conversely, the transcription of PINs that stabilizes the position of the stem cell niche is maintained by PLTs [10,42,48]. Moreover, on the fifth day of root growth, maximal meristem size and stabilization of the transition zone are established by the auxin signaling via *PLT* action/cytokinin signaling network [49].

Both GCN5 and ADA2b promote the expression of *PIN1*, *PIN3* and *PIN4* to higher levels necessary for the physiological needs of the growing root. ADA2b has been shown to interact with transcription factors involved in auxin responses, recruiting GCN5 to the genomic regions needed [25,50]. In addition, ADA2b is required for histone acetylation in response to auxin [50,51]. The expression of *PIN4* was upregulated after treatment with trichostatin A (TSA), a potent histone deacetylase inhibitor, suggesting that this gene could be regulated by histone acetylation [50]. These data suggest that *PIN4* induction requires GCN5-specific histone acetylation. In our results, *PIN4* is one of the genes affected by GCN5 and ADA2b since its expression is reduced in mutants.

Interestingly, *PIN4* shows reduced histone H3K14 acetylation in both the promoter and the coding region, compared to *PIN1* and *PIN3*, which only show a reduction in the coding region in *gcn5* mutants. Therefore, *PIN4* needs histone acetylation in its genomic locus for proper induction, and this acetylation could be accomplished by GCN5. In our results, GCN5 but not ADA2b affected the H3K14 acetylation at *PIN1, 3* and *4* loci, although all three genes are downregulated in *gcn5* and *ada2b* mutants. Therefore, GCN5-dependent histone acetylation is required to adequately express genes involved in auxin transport in the early days of root growth.

Maximal meristem size and stabilization of the transition zone are established by the auxin/PLT/ARR-B network five days after germination [49]. GCN5 and ADA2b also affect cytokinin levels by regulating cytokinin biosynthesis and catabolism [23]. Furthermore, auxin-induced cytokinin signaling depends on GCN5-ADA2b action [23]. Cytokinins were shown to control auxin transport, unitizing transcriptional and post-transcriptional regulation of PIN efflux carriers, thus controlling intercellular auxin distribution in the root tips [52,53,54]. Our results suggest that the synergistic network of auxin and cytokinin signaling requires a GCN5 and ADA2b complex that modulates histone acetylation of auxin efflux transporters during the early days of root growth. Moreover, it indicates an essential role of GCN5-dependent histone acetylation in the interaction between auxin transport, auxin signaling, cytokinin signaling and regulatory circuits during root growth. Root meristem size is in a perpetual change in response to external and internal conditions, signaled by the changing levels of histone acetylation and other chromatin modifications to modulate hormone responses. Integrating of all this information to alter root growth dynamically and the coordination required across the whole root system is still an open question for further studies.

## 4. Conclusions

In this manuscript we describe a biological role for GCN5 histone acetyltransferase and ADA2b transcriptional adaptor, during early root growth. Both proteins promote the expression of PIN1, PIN3 and PIN4 auxin transporters of the PIN family and help maintain a proper auxin distribution for normal root growth. These results place GCN5 and ADA2b as important players in the regulatory module that maintains an active root, affecting both auxin and cytokinin responses as well as their intertwined network of interactions. As root meristems integrate internal and external cues to guide root growth in the most effective way for the plant, factors that affect their function are of the utmost importance. Since the study of root growth and development is challenging in economically important crops and even more so in tree species, the study of model organisms such as *Arabidopsis thaliana* provides valuable insights on the hormonal and molecular mechanisms that could find their way into horticultural practice.

## 5. Materials and Methods

### 5.1. Plant Materials and Growth

The *Arabidopsis thaliana* (L) Heynh. *gcn5-1* and *ada2b-1* mutants in Wassilewskija-2 (Ws-2) background were acquired from the University of Wisconsin Arabidopsis Knockout Facility as previously described in [20], and were available to the lab. The PIN1:PIN1-GFP reporter line, was acquired by NASC (N23889). The PIN1:PIN1-GFP *gcn5-1/+* line was created by crossing the PIN1:PIN1-GFP line with the *gcn5-1* mutant as described by [31], and was available to the lab. Commercially available bleach was used for the sterilization of the seed surface. Seeds were stratified at 4 °C for 3–4 days in the dark. For plating, the Gamborg B5 (GB5) medium, including vitamins (Duchefa Biochemie, Amsterdam, Netherlands) supplemented with 1% sucrose (Duchefa) and 0.8% phytoagar (Duchefa) and adjusted at pH 5.7, was used. Plants were cultivated at an average of 22 °C with 70–80 mol m^−2^ s^−1^ cool-white, fluorescent lamps in long days (16 h light/8 h dark). Commercially available soil, Terrahum^®^ (Deutsche Kompost Handelsgesellschaft), was used for cultivation.

### 5.2. Genetic Analysis and Genotyping

The PIN1:PIN1-GFP *ada2b-1/+* line was created using PIN1:PIN1-GFP pollen to fertilize *ada2b-1/+* gynoecia. The F1 was left to self-fertilize, and the various genotypes were identified in the F2 using a combination of phenotypic and PCR-based methods (Appendix A for primers). The *gcn5-1* and *ada2b-1* mutant alleles were tracked using PCR with specific primers.

### 5.3. Gene Expression Analysis

Ws-2, *gcn5-1* and *ada2b-1* plants were grown vertically in GB5 containing-medium plates for seven days for gene expression assays. Whole roots were collected and flash-frozen in liquid nitrogen, and RNA was extracted using the Nucleospin^®^ RNA Plant kit (Macherey- Nagel, Duren, Germany). In at least three independent biological repeats, reverse transcription was performed using 0.5 μg of total RNA with the PrimeScript^TM^ first strand cDNA Synthesis kit (TaKaRa, Shiga, Japan). Quantitative reverse-transcription polymerase chain reactions (RT-qPCRs) were prepared with the AMPLIFYME SG Universal Mix (AM02) (BLIRT SA, Gdansk, Poland) at the ABI StepOne™ system (Applied Biosystems, Foster City, CA, USA). Three technical repeats were run for each sample. The *At4g26410* gene was used as an endogenous control (Appendix A). Data were analyzed with the ΔΔCt method using StepOne Software 2.1. Statistical significance was calculated using a one-way ANOVA, with Fisher’s Least Significant Difference (LSD) and post-hoc test with a 95% confidence interval using IBM SPSS Statistics software 23.0 (Statistical Product and Service Solutions, USA).

### 5.4. Microscopy

A Zeiss AxioImager.Z2 (Carl Zeiss AG, Munich, Germany) equipped with a digital AxioCam MRc 5 camera for green fluorescent protein (GFP) detection was used for most observations. At a minimum, 30 seedlings were observed on 4, 5 and 6 days, and 80 were observed on the seventh day. The integrated intensity of GFP was measured by ImageJ software, and values were normalized as previously described [55]. Statistical significance was calculated using an independent samples *t*-test, with a 95% confidence interval, using IBM SPSS Statistics software 23.0 (Statistical Product and Service Solutions, USA).

For confocal microscopy, seven-day-old seedlings were stained with 10 μg/mL propidium iodide (PI) (Sigma-Aldrich, St Louis, MO, USA) and mounted directly in the PI solution. Root caps were visualized under a Zeiss Observer.Z1 microscope (Carl Zeiss AG, Munich, Germany), equipped with an LSM780 confocal laser scanning module to detect GFP. Imaging was achieved with ZEN2011 software according to the manufacturer’s instructions.

### 5.5. Chromatin Immunoprecipitation (ChIP)

As previously described, a ChIP assay was performed with minor modifications [23,56]. Seven-day-old seedlings were harvested after selection, and a total of approximately 250, 650, and 900 roots for wild-type Ws-2, *gcn5-1*, and *ada2b-1*, respectively, were collected. The *ada2b-1* roots were selected from a segregating population of more than 4000 seedlings. The total amount of tissue used for this experiment was 25 mg per sample. The exact amount of tissue was used for all three genotypes. Antibodies against acetylated histone H3K14 (Anti-Histone H3 (Lys14), EMD Millipore #07-353, Burlington, MA, USA) and non-acetylated H3 (ChIPAb + Histone H3 C-term, EMD Millipore #17-10046) were used. Immunoprecipitated DNA was diluted and analyzed by RT-qPCR (Appendix A for primers used). RT-qPCRs were performed using the AMPLIFYME SG Universal Mix (AM02) (BLIRT SA, Gdánsk, Poland) at the ABI StepOne™system (Applied Biosystems, Foster City, CA, USA). Samples were analyzed and calibrated as previously described [20,23]. Statistical significance was calculated using a one-way ANOVA, with Fisher’s Least Significant Difference (LSD) and post-hoc test with a 95% confidence interval using IBM SPSS Statistics software (Statistical Product and Service Solutions, USA).

## Figures and Tables

**Figure 1 plants-11-03572-f001:**
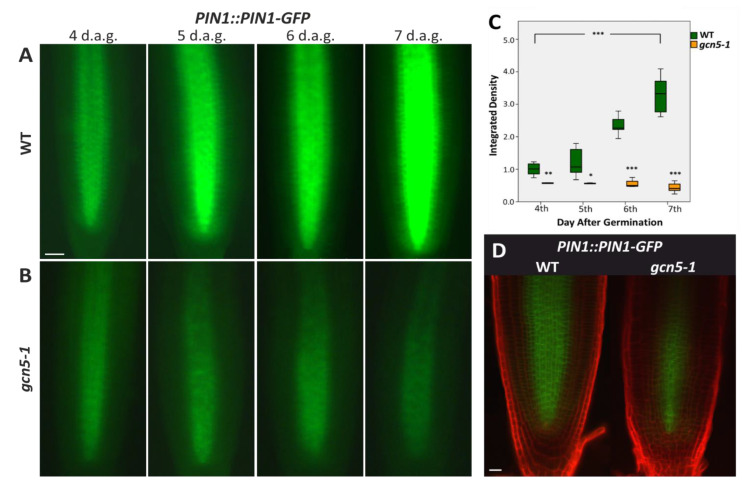
*GCN5* affects PIN1 expression during early root growth in *Arabidopsis thaliana*. The expression of reporter gene PIN1::PIN1-GFP in roots of (**A**) wild type (WT) and (**B**) *gcn5-1* mutant plants at the fourth, fifth, sixth and seventh day after germination (d.a.g.). Scale bars represent 50 μm. (**C**) Graphical representation of fluorescence density between WT and *gcn5-1*. The bars represent the range of the two quadrants. The horizontal line in the bar is the median, while the terminals are the minimum and maximum values of the data. Asterisks above bars of *gcn5-1* indicate statistical significance compared to the same d.a.g. of WT. Asterisks between brackets show the difference between fourth and the seventh d.a.g., using an independent samples *t*-test: * *p* < 0.05, ** *p* < 0.01, and *** *p* < 0.001. (**D**) Confocal microscopy for PIN1::PIN1-GFP expression in 7-day-old roots of WT and *gcn5-1* after PI staining. Scale bars are 20 μm.

**Figure 2 plants-11-03572-f002:**
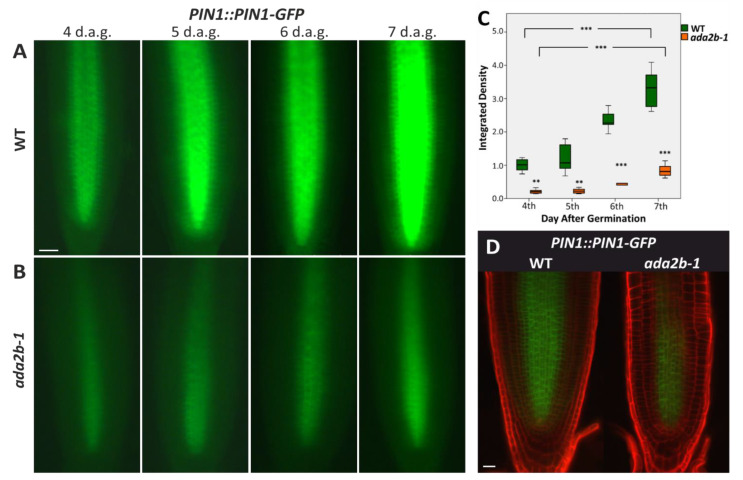
*ADA2b* affects PIN1 expression during early root growth in *Arabidopsis thaliana*. The expression of reporter gene PIN1::PIN1-GFP in roots of (**A**) wild type (WT) and (**B**) *ada2b-1* mutant plants at the fourth, fifth, sixth and seventh day after germination (d.a.g.). Scale bars represent 50 μm. (**C**) Graphical representation of fluorescence density between WT and *ada2b-1*. The bars represent the range of the two quadrants. The horizontal line in the bar is the median, while the terminals are the minimum and maximum values of the data. Asterisks above bars of *ada2b-1* indicate statistical significance compared to the same d.a.g. of WT. Asterisks between brackets show the difference between the fourth and seventh d.a.g., using an independent samples *t*-test: ** *p* < 0.01, and *** *p* < 0.001. (**D**) Confocal microscopy for PIN1::PIN1-GFP expression in 7-day-old roots of WT and *ada2b-1* after PI staining. Scale bars are 20 μm.

**Figure 3 plants-11-03572-f003:**
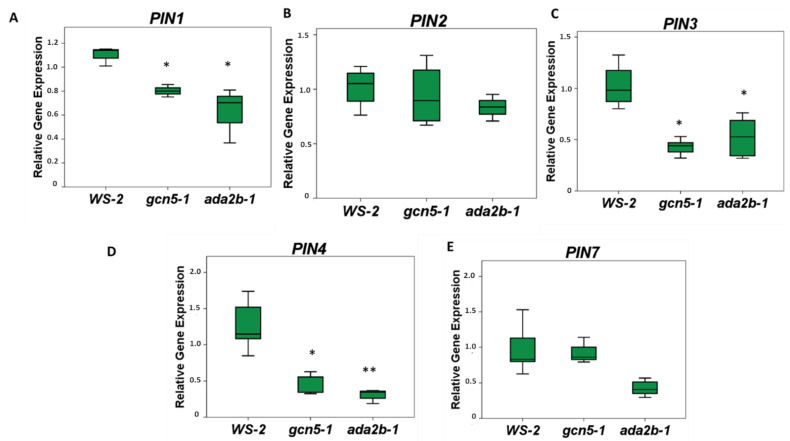
The role of GCN5 and ADA2b in *PIN* gene expression in *Arabidopsis thaliana* roots. The roots of seven-day-old Ws-2, *gcn5-1*, and *ada2b-1* seedlings were harvested for qRT-PCR analysis for the genes (A) *PIN1*, (B) *PIN2*, (C) *PIN3*, (D) *PIN4*, and (E) *PIN7*. The bars represent the range of the two quadrants. The horizontal line in the bar is the median, while the terminals are the minimum and maximum values of the data. Asterisks indicate the statistical significance based on an independent samples *t*-test: * *p* < 0.05, and ** *p* < 0.01.

**Figure 4 plants-11-03572-f004:**
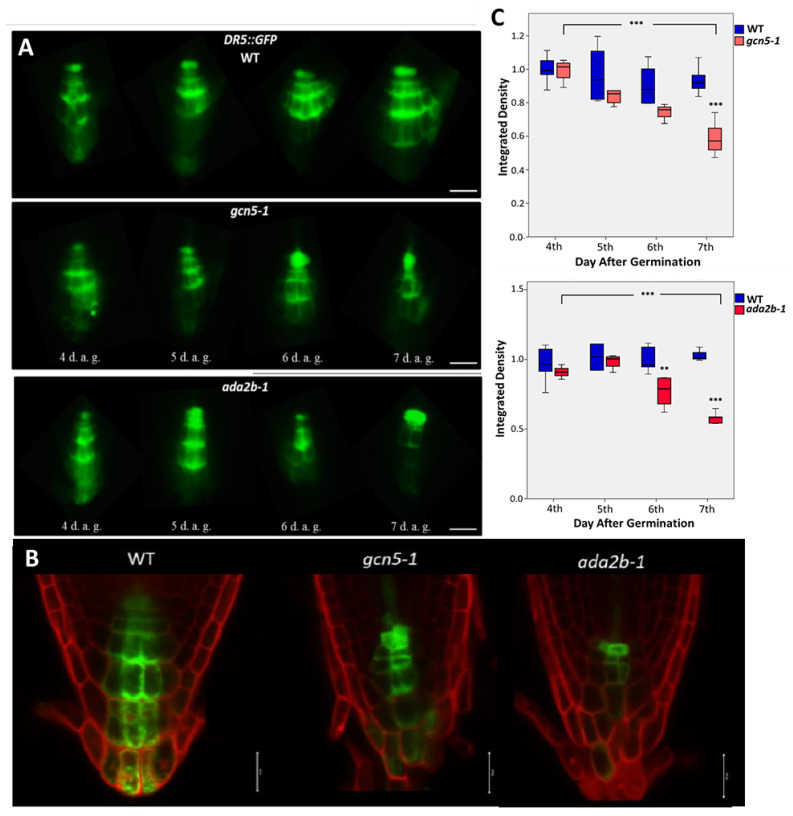
GCN5 and ADA2b affect auxin distribution during early root growth. (**A**) The expression of the auxin reporter DR5rev::GFP in roots of wild type, *gcn5-1* and *ada2b-1* mutant plants at the fourth, fifth, sixth, and seventh day after germination (d.a.g.). Scale bars represent 50 μm. (**B**) Confocal microscopy for DR5rev::GFP expression in 7-day-old roots of WT, *gcn5-1* and *ada2b-1* after PI staining. Scale bars are 20 μm. (**C**) Fluorescence density between wild-type and *gcn5-1* (upper graph) and wild-type and *ada2b-1* (lower graph) during root growth. The bars represent the range of the two quadrants. The horizontal line in the bar is the median, while the terminals are the minimum and maximum values of the data. Asterisks above bars of *ada2b-1* indicate statistical significance compared to the same d.a.g. of wild-type. Asterisks between brackets show the difference between 4th and seventh d.a.g., using an independent samples *t*-test: ** *p* < 0.01, and *** *p* < 0.001.

**Figure 5 plants-11-03572-f005:**
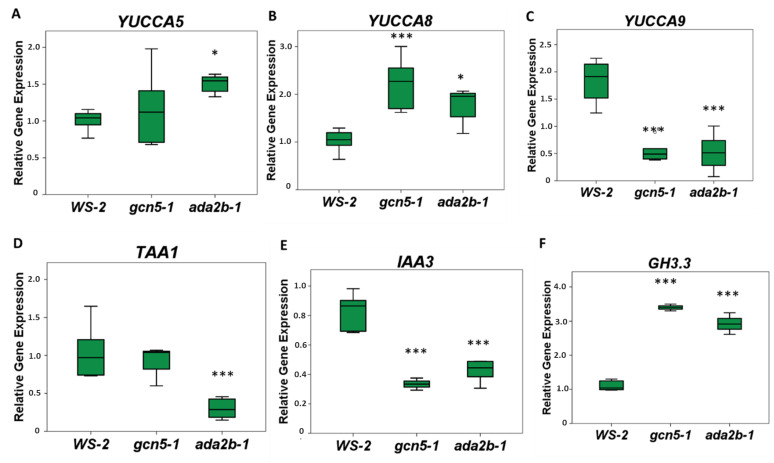
The role of GCN5 and ADA2b in *auxin-related* genes in *Arabidopsis thaliana* roots. The roots of seven-day-old Ws-2, *gcn5-1*, and *ada2b-1* seedlings were harvested for qRT-PCR analysis for the genes (**A**) *YUCCA 5*, (**B**) *YUCCA 8*, (**C**) *YUCCA 9*, (**D**) *TAA1*, (**E**) *IAA3* and (**F**) GH3.3. The bars represent the range of the two quadrants. The horizontal line in the bar is the median, while the terminals are the minimum and maximum values of the data. Asterisks indicate the statistical significance based on an independent samples *t*-test: * *p* < 0.05 and *** *p* < 0.001.

**Figure 6 plants-11-03572-f006:**
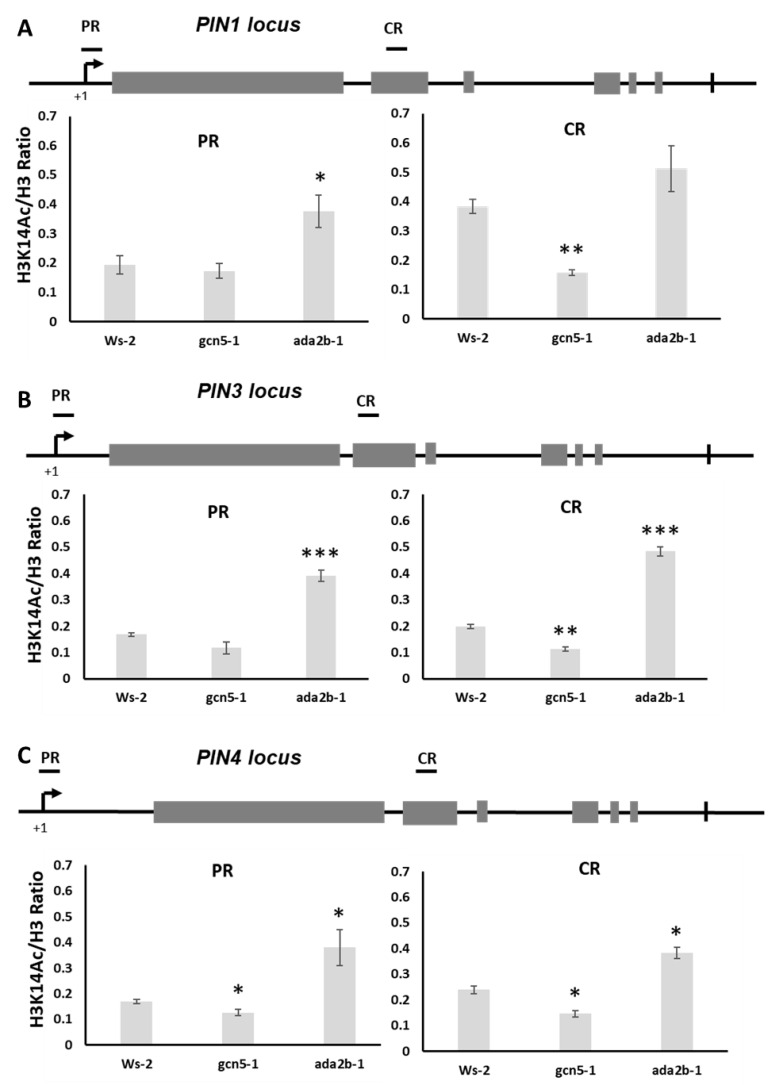
GCN5 and ADA2b affect histone acetylation of H3K14 of PIN genes. ChIP assays in roots of seven-day-old Ws-2, *gcn5-1*, and *ada2b-1* plants. The promoter (P) and the coding region (CR) were analyzed for each genomic locus as indicated by the line above the gene model. (**A**) PIN1, (**B**) *PIN3*, and (**C**) *PIN4* loci were analyzed. Values are presented as a percentage of input. Asterisks indicate the statistical significance of three technical repeats compared to the wild-type, based on an independent samples *t*-test: * *p* < 0.05, ** *p* < 0.01, and *** *p* < 0.001.

## Data Availability

The data that support the findings of this study are available from the corresponding author upon reasonable request.

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
