# Peer review of "Histone Acetyltransferase GCN5 Affects Auxin Transport during Root Growth by Modulating Histone Acetylation and Gene Expression of PINs"

_plants, 2022, doi:10.3390/plants11243572_

Round 1
Reviewer 1 Report
The manuscript addresses an important issue of plant root growth regulation through auxin in Arapidopsis model plants. The experimental design is appropriate and the manuscript is well organized and presented. However, there are a couple of issues that the authors should elaborate to improve readership and clarity of the manuscript. These are as follows:
1. The introduction should be supplemented with a statement regarding why is so important to decipher auxin-regulated root growth development in the model plant Arabidopsis as well as in other plant species, including trees as part of the Special Issue titled “ Role of Gene and Hormone Regulation in Adventitious Root Formation in Trees”.
1. In the Introduction section a short paragraph regarding the role of auxin, indicating IAA , in root growth and development regulation should be added including recent research article(s) and/or review (i.e. doi: 10.3390/ijms19113656) to provide a general sight of the current knowledge.
2. There are a number of self-citations, specifically 8 out of 52 total references. This should be limited using review articles (i.e. doi.org/10.1104/pp.17.00765).
3. In the Results section 2.1. is stated ”However, the pattern of PIN1 expression remains similar to wild-type plants (Fig.1D)”. I argue that the expression pattern of PIN1 is not similar to the wild-type (Fig. 1A, B). In Fig 1D the expression of PIN1:PIN1-GFP is reduced in gcn5-1 mutant line compared to control with time (d) as depicted also in (Fig 1B). As the expression of PIN1:PIN1-GFP in gcn5-1 mutant background is dwindling with time (Fig 1B), then the conclusion that GCN5 has a synergistic positive effect on PIN1 expression could be stated. Please elaborate appropriate rephrasing.
4. In the Discussion section is stated “More specifically, it is possible that there are auxin degradation mechanisms, which in the absence of GCN5 and ADA2b are overproduced.” The experimental design and results presented indicate that beyond GCN5 and ADA2b there are other component(s) implicated in auxin biosynthesis and regulation that await to be unveiled. Thus in the absence of both GCN5 and ADA2b auxin degradation is promoted. The statement should be rephrased accordingly.
Author Response
Respond to Reviewer 1 for Poulios et al, Histone acetyltransferase GCN5 affect auxin transport during root growth by modulating histone acetylation and gene ex-pression of PINs
The manuscript addresses an important issue of plant root growth regulation through auxin in Arapidopsis model plants. The experimental design is appropriate and the manuscript is well organized and presented. However, there are a couple of issues that the authors should elaborate to improve readership and clarity of the manuscript. These are as follows:
We would like to thank the reviewer for the fruitful comments. Below are our responses to the requests.
The introduction should be supplemented with a statement regarding why is so important to decipher auxin-regulated root growth development in the model plant Arabidopsis as well as in other plant species, including trees as part of the Special Issue titled “ Role of Gene and Hormone Regulation in Adventitious Root Formation in Trees”.
We add a paragraph to address this request. Please check the revised version, ln 92 to 96. The references were also updated.
In the Introduction section a short paragraph regarding the role of auxin, indicating IAA , in root growth and development regulation should be added including recent research article(s) and/or review (i.e. doi: 10.3390/ijms19113656) to provide a general sight of the current knowledge.
We add a paragraph to address this request. Please check the revised version, ln 62 to 73. The references were also updated.
There are a number of self-citations, specifically 8 out of 52 total references. This should be limited using review articles (i.e. doi.org/10.1104/pp.17.00765).
Thank you for the suggestion but those references highlight the importance of GCN5 in plant development, therefore we decided to keep them.
In the Results section 2.1. is stated ”However, the pattern of PIN1 expression remains similar to wild-type plants (Fig.1D)”. I argue that the expression pattern of PIN1 is not similar to the wild-type (Fig. 1A, B). In Fig 1D the expression of PIN1:PIN1-GFP is reduced in gcn5-1 mutant line compared to control with time (d) as depicted also in (Fig 1B). As the expression of PIN1:PIN1-GFP in gcn5-1 mutant background is dwindling with time (Fig 1B), then the conclusion that GCN5 has a synergistic positive effect on PIN1 expression could be stated. Please elaborate appropriate rephrasing.
Thank you for the suggestion, we have rephrased the paragraph. Please check the revised version, ln 131 to 136.
In the Discussion section is stated “More specifically, it is possible that there are auxin degradation mechanisms, which in the absence of GCN5 and ADA2b are overproduced.” The experimental design and results presented indicate that beyond GCN5 and ADA2b there are other component(s) implicated in auxin biosynthesis and regulation that await to be unveiled. Thus in the absence of both GCN5 and ADA2b auxin degradation is promoted. The statement should be rephrased accordingly.
Thank you for the suggestion, we have rephrased the paragraph. Please check the revised version, ln 297 to 302.

Reviewer 2 Report
The MS entitled “Histone acetyltransferase GCN5 affect auxin transport during root growth by modulating histone acetylation and gene expression of PINs” is written well and data are quite interesting.
Such as SAGA? Not clear Kindly re-write again
I will suggest to authors please make a list of abbreviations after the abstract section for ADA2b, PIN1. GFP, GCN etc…………………. It would be good for the readers.
P.11: L 312 : The Arabidopsis thaliana (L) Heynh. gcn5-1 and ada2b-1 mutants in Wassilewskija-2 (Ws-2) background were previously described in [13]?????????????????? Not clear
The PIN1:PIN1-GFP gcn5-1/+ line 313 was described by [26].?????????????
Commercially available bleach was used for the sterilization of the 314 seed surface. Seeds were stratified at 4οC for 3–4 days in the dark.- re-write, “ 4°c”
22–24 οC????????????????
I could not see the Supplementary Materials:
Results:
Therefore, we “ ask” if GCN5 could also affect PIN1 expression during root growth????????? Re-write again
Discussion: First, new auxin molecules are not biosynthesized in the cells produced during root 237 growth: evidences required??????????????
Conclusion part is missing, Author’s must provide the translational value of their study.
Author Response
Response to reviewer 2
The MS entitled “Histone acetyltransferase GCN5 affect auxin transport during root growth by modulating histone acetylation and gene expression of PINs” is written well and data are quite interesting.
We would like to thank the reviewer for the comments. Below are our response to the requests.
Such as SAGA? Not clear Kindly re-write again
We include the names in the first mention abbreviation.
I will suggest to authors please make a list of abbreviations after the abstract section for ADA2b, PIN1. GFP, GCN etc…………………. It would be good for the readers.
We add a list of abbreviations after the abstract
P.11: L 312 : The Arabidopsis thaliana (L) Heynh. gcn5-1 and ada2b-1 mutants in Wassilewskija-2 (Ws-2) background were previously described in [13]?????????????????? Not clear
The PIN1:PIN1-GFP gcn5-1/+ line 313 was described by [26].?????????????
We rephrased the paragraph. Please check the revised form, ln 367-372.
Commercially available bleach was used for the sterilization of the 314 seed surface. Seeds were stratified at 4οC for 3–4 days in the dark.- re-write, “ 4°c”
22–24 οC????????????????
We corrected them accordingly
I could not see the Supplementary Materials:
We add the supplementary table
Results:
Therefore, we “ ask” if GCN5 could also affect PIN1 expression during root growth????????? Re-write again
We rephrase the paragraph. Please check the revised form, ln 103-111.
Discussion: First, new auxin molecules are not biosynthesized in the cells produced during root 237 growth: evidences required??????????????
We rephrase the paragraph. Please check the revised form, ln 275-278.
Conclusion part is missing, Author’s must provide the translational value of their study.
Thank you for the reminder. The conclusion section was added in ln 351-363
Round 2
Reviewer 2 Report
The Ms has been revised accordingly.